# HPOBench: A Collection of Reproducible Multi-Fidelity Benchmark Problems for HPO

**Katharina Eggensperger**[1][*]**, Philipp Müller**[1]**, Neeratyoy Mallik**[1]**, Matthias Feurer**[1]**,
René Sass**[2]**, Aaron Klein**[3][†]**, Noor Awad**[1]**, Marius Lindauer**[2]**, Frank Hutter**[1,4]
[1] Albert-Ludwigs-Universität Freiburg [2] Leibniz Universität Hannover
[3] Amazon [4] Bosch Center for Artificial Intelligence

## Abstract

To achieve peak predictive performance, hyperparameter optimization (HPO) is a crucial component of machine learning and its applications. Over the last years, the number of efficient algorithms and tools for HPO grew substantially. At the same time, the community is still lacking realistic, diverse, computationally cheap, and standardized benchmarks. This is especially the case for multi-fidelity HPO methods. To close this gap, we propose *HPOBench*, which includes 7 existing and 5 new benchmark families, with a total of more than 100 multi-fidelity benchmark problems. *HPOBench* allows to run this extendable set of multi-fidelity HPO benchmarks in a reproducible way by isolating and packaging the individual benchmarks in containers. It also provides surrogate and tabular benchmarks for computationally affordable yet statistically sound evaluations. To demonstrate *HPOBench*'s broad compatibility with various optimization tools, as well as its usefulness, we conduct an exemplary large-scale study evaluating 13 optimizers from 6 optimization tools. We provide *HPOBench* here: `https://github.com/automl/HPOBench`.

## 1 Introduction

The plethora of design choices in modern machine learning (ML) makes research on practical and effective methods for hyperparameter optimization (HPO) ever more important. In particular, ever-growing models and datasets create a demand for new HPO methods that are more efficient and powerful than existing black-box optimization (BBO) methods. Especially if it is only feasible to evaluate very few models fully, multi-fidelity optimization methods have been shown to yield impressive results by trading off cheap-to-evaluate proxies and expensive evaluations on the real target [1–5]. They showed tremendous speedups, such as accelerating the search process in low-dimensional ML hyperparameter spaces by a factor of 10 to 1000 [2, 5]. However, the development of such methods often happens in isolation, which potentially prevents HPO research from reaching its full potential. Prior publications on new HPO methods (i) often relied on artificial test functions and low-dimensional toy problems, (ii) sometimes introduced a new set of problems, (iii) set up on different computing environments, having different requirements and interfaces, and (iv) often did not open-source their code base. All of these make it difficult to compare and develop methods, necessitating an evolving set of relevant and up-to-date benchmark problems which drives continued and quantifiable progress in the community.

While there are efforts to simplify benchmarking HPO and global optimization algorithms [6–12], we are not aware of efforts to collect a diverse set of benchmarks in a single library, with a unified

---

[*]{eggenspk,mallik,fh}@cs.uni-freiburg.de
[†]work done prior to joining Amazon

35th Conference on Neural Information Processing Systems (NeurIPS 2021) Track on Datasets and Benchmarks.

interface and countering potentially conflicting dependencies that may arise over time. The latter is particularly important because the rapid evolution of the Python-ML ecosystem can render a benchmark no longer usable for the community after a major release was published. This creates a significant hurdle for contribution from the community to grow a benchmark library. To solve this issue, we propose *HPOBench*, a benchmark suite for HPO problems, with a special focus on multi-fidelity problems, licensed under a permissive OSS license (*Apache 2.0*) and available at `https://github.com/automl/HPOBench`. *HPOBench* provides a common interface and an infrastructure to isolate benchmarks in their own containers and implements 12 popular benchmark families, each with multiple problems and preserved with its dependencies in a container for long-term use. To enable efficient comparisons, most of these benchmarks are table- or surrogate-based, enabling resource efficient large-scale experiments, which we demonstrate in this work. Our contributions are:

1. The first available collection of multi-fidelity HPO problems. It contains 12 benchmark families with 100+ multi-fidelity HPO problems under a unified interface, comprising traditional HPO and neural architecture search (NAS). These benchmarks also define the largest collection of black-box HPO problems to date.

2. The first collection of *containerized* benchmarks to ensure the longevity, maintainability and extensibility of benchmarks.

3. The first set of HPO benchmarks that are available as both, the *raw* benchmark and the *tabular* version.

4. The first HPO benchmark that also supports multi-objective optimization and transfer-HPO across datasets (and arbitrary combinations of these with multiple fidelities).

5. We demonstrate how *HPOBench* can be used in an exemplary large-scale study with 13 optimizers from 6 optimization tools, assessing whether advanced methods outperform random search and how effective multi-fidelity HPO is.

This paper is structured as follows. We first discuss background on HPO and multi-fidelity optimization (Section 2). Then, we discuss related work on benchmarking (Section 3). Next, we describe the challenges for an HPO benchmark and how *HPOBench* alleviates them (Section 4). Then, we conduct a large-scale comparison of existing, popular HPO methods to demonstrate the usefulness of *HPOBench* (in Section 5). We conclude the paper by highlighting further advantages and potential future work (Section 6).

## 2  Background on Hyperparameter Optimization

With *HPOBench* we aim to provide benchmarks to evaluate HPO methods. In the following, we briefly formalize BBO for HPO and survey multi-fidelity optimization (see Feurer and Hutter [13] for a detailed overview), both with a focus on the methods used in our experiments.

### 2.1  Black-box Hyperparameter Optimization

Black-box optimization (BBO) aims to find a solution $\arg\min_{\boldsymbol{\lambda}\in\boldsymbol{\Lambda}} f(\boldsymbol{\lambda})$ where $f$ is a black-box function, for which typically no gradients are available, we cannot make any statements about its smoothness, convexity and noise level. In summary, the only mode of interaction with black-box functions is querying them at given inputs $\boldsymbol{\lambda}$ and measuring the quantity of interest $f(\boldsymbol{\lambda})$. In the context of HPO, $\boldsymbol{\lambda} \in \boldsymbol{\Lambda}$ is a hyperparameter configuration where the domain $\Lambda_i$ of a hyperparameter is often bounded and continuous, but can also be integer, ordinal or categorical. There are also so-called conditional hyperparameters [14, 15] defining hierarchical search spaces; however, the first version of *HPOBench* focuses on flat configuration spaces first as all optimizers support this.

There are three broad families of BBO methods: (i) purely explorative approaches such as Random Search (*RS*) and grid search are simple but sample-inefficient; (ii) model-free Evolutionary Algorithms (EAs) based on mutation, crossover and selection operators applied to a population of configurations require comparably large resources to evaluate the entire population but can perform very well given enough resources; (iii) iterative model-based methods, such as Bayesian Optimization [16], which are guided by a predictive model trained on prior function evaluations are known as the most sample-efficient methods. We include representative algorithms from each of these 3 families in our exemplary experiments in Section 5.

## 2.2 Multi-fidelity Hyperparameter Optimization

To efficiently optimize today's ever-growing ML models, multi-fidelity approaches relax the black-box assumption by allowing cheaper queries at lower fidelities $b$ as well ($\arg\min_{\boldsymbol{\lambda}\in\boldsymbol{\Lambda}} f(\boldsymbol{\lambda}, b)$). Examples for these approximations include dataset subsets [2, 17, 18], feature subsets [19] or lower number of epochs [19–21]. Multi-fidelity methods have been shown to lead to speedups of up to $1000\times$ over black-box methods [2, 5]. *HPOBench* will allow the community to compare different multi-fidelity methods and in the following we give an overview of representative methods.

A popular multi-fidelity HPO approach that discretizes the fidelity space is Hyperband (*HB* [19]), a very simple method with strong empirical performance. It randomly samples new configurations and allocates more resources to promising configurations by repeatedly calling successive halving (*SH* [4]) as a sub-algorithm. The simplicity and effectiveness of *HB* have been leveraged with other popular black-box optimizers for improved performance: *BOHB* [22] combines *HB* with Bayesian Optimization (*BO*) and DEHB [5] combines it with the evolutionary approach of Differential Evolution (*DE* [23, 24]). The non-*HB*-based multi-fidelity case has also been researched extensively [2, 3, 18, 20, 21, 25–28]. Not being limited to predefined fidelity values makes these methods very powerful, but they rely on strong models to avoid poor choices of fidelities, often making HB-based fidelity selection more robust. To study the efficacy of multi-fidelity optimization, in our exemplary experiments in Section 5, we primarily compared black-box optimizers against their multi-fidelity versions (i.e., *RS* vs. *HB*, *BO* vs. *BOHB*, and *DE* vs. *DEHB*). These experiments show large speedups of multi-fidelity optimizers in the regime of small compute budgets, whereas for large compute budgets multi-fidelity optimization is less useful.

Besides multi-fidelity optimization, a very active field of study to speed up HPO is to use transfer-learning across datasets [29–33]; we note that transfer HPO methods can also be evaluated with *HPOBench* by learning across the datasets within each of its families.

## 3 Related Work

Proper benchmarking is hard. It is important to be aware of technical and methodological pitfalls, e.g. comparing implementations instead of algorithms [34, 35], comparing tuned algorithms versus untuned baselines [36, 37], to not fall for an illusion of progress [38, 39] and to know which sources of variance exist and control for them [40]. Also, there is a rich literature on how to empirically evaluate and compare methods in various domains, e.g. evolutionary optimization [41], planning [42], satisfiability and constraint satisfaction [43], algorithm configuration [44], NAS [45], and also for benchmarking optimization algorithms [46]. Our goal is not to provide further recommendations on how and why to benchmark, but to provide concrete benchmarks to simplify development and to improve the reproducibility and comparability of HPO and in particular multi-fidelity methods.

Furthermore, there have been a lot of efforts to provide optimization benchmarks for the community. Having a common set of benchmark problems in a unified format fosters and guides research. Prominent examples in the area of HPO are ACLib [47] for algorithm configuration, COCO [9] for continuous optimization, Bayesmark [8] for Bayesian optimization, Olympus [12] for optimization of experiment planning tasks, and HPO-B [48] for transfer-HPO methods (for more, see Appendix B). However, no benchmark so far has multi-fidelity optimization problems, supports preserving a diverse set of benchmarks for the longer term (containers), supports multiple objectives, and provides cheap-to-evaluate surrogate/tabular benchmarks; we hope to close this gap with *HPOBench*.

Besides benchmarks, competitions are another form of focusing research effort by providing a common goal and incentive. Famous examples are the *AutoML* challenges [49], the *AutoDL* challenge [50], the GECCO BBOB workshop series based on COCO [9] and the NeurIPS 2020 BBO challenge [51] (for more, see Appendix C). In contrast to these, we do not focus on defining concrete experimentation protocols, but rather on providing a flexible benchmarking environment to study, develop and compare optimization methods.

## 4 HPOBench: A Benchmark Suite for Multi-Fidelity Hyperparameter Optimization benchmarks

In this section, we present *HPOBench*, a collection of HPO benchmarks defined as follows:

**Definition 1 (HPO Benchmark)** *An HPO benchmark consists of a function $f : \lambda \to \mathcal{R}$ to be minimized and a (bounded) hyperparameter space $\Lambda$ with hyperparameters $[\Lambda_1, \ldots, \Lambda_d]$ of type continuous, integer, categorical or ordinal. In the case of multi-fidelity benchmarks, $f$ can be queried at lower fidelities, $f : \lambda \times b \to \mathcal{R}$, and the fidelity space $B$ describes which low-fidelities $[B_1, \ldots, B_e]$ of type continuous, integer or ordinal are available.*

Specifically, each benchmark consists of the implementation of that function, which returns at least one loss. Since this function typically evaluates an ML algorithm, the benchmark defines all relevant settings, dependencies and inputs, such as datasets, splits and how to compute the loss.

In the remainder of this section, we first discuss the desiderata of a benchmark that aids HPO research and then highlight the features of *HPOBench* by detailing how its design fulfills these desiderata.

## 4.1 Desiderata of an HPO Benchmark

One of the challenges posed to standardized HPO research lies in the varied choices of the underlying ML components – datasets and their splits, preprocessing, hyperparameter ranges, underlying software versions, and hardware used. Moreover, the practices applied in HPO research itself can vary along the lines of optimization budget, number of repetitions, metrics measured and reported. This leads to inconsistencies and difficulties in comparison of different HPO methods across publications and over time, affecting the reproducibility of experiments that hinders continued progress in HPO research.

In order to alleviate such issues and encourage participation by the research community, benchmarks need to standardize these practices to allow the community to be an active stakeholder in developing and re-using benchmarks. *HPOBench* is designed to both allow easy, flexible use with a minimal API that is identical for all benchmarks (see Figure 2); and have a low barrier for contributing new benchmark problems. We, therefore, identify 3 features of a benchmark that allow its wide-scale use and long-term applicability: (i) *efficiency* by providing tabular and surrogate benchmarks for quick, efficient experiments, along with the original benchmarks; (ii) *reproducibility* of results by containerizing benchmarks; and (iii) *flexibility* by covering different optimization landscapes and possible use cases, e.g. multi-objective, transfer-HPO, and even multi-fidelity optimization with multiple fidelity variables. To our knowledge, no other existing benchmarks offer these possibilities. *HPOBench* provides a framework to enable standardized, principled research and experimentation. We list all benchmarks that are included in *HPOBench* in Table 1 and provide a detailed description of the respective configuration spaces in Appendix D.

## 4.2 Efficiency

HPO benchmarks that follow Definition 1 exhibit the drawback that they evaluate a costly function, rendering the empirical comparison of optimization algorithms expensive and ruling out such benchmarks for interactive development of new methods. To overcome this issue, beside such raw benchmarks, we also provide two well-established benchmark classes which alleviate this issue:

**Definition 2 (Tabular Benchmark)** *A tabular benchmark returns values from a lookup table with recorded function values of a raw HPO benchmark instead of evaluating $f(\lambda)$. The (bounded) hyperparameter space is restricted to only contain these values and therefore bears a form of discretization. In the case of multi-fidelity benchmarks, each tabular benchmark has a fidelity space and the underlying table also contains the recorded function values on the low-fidelities.*

Tabular benchmarks are popular in the HPO community as they are easy to distribute and induce little overhead [52, 30, 53–55], however, they require to discretize the hyperparameter space. Surrogate benchmarks [56, 57] are an alternative since they provide the original hyperparameter space.

**Definition 3 (Surrogate Benchmark)** *A surrogate benchmark returns function values predicted by an ML model trained on a tabular benchmark or recorded function values of a raw HPO benchmark. It reuses the original hyperparameter space and can be extended to the multi-fidelity case as well.*

While surrogate benchmarks are similarly cheap to query, the surrogate's internal ML model adds extra complexity and the benchmark's quality crucially depends on the quality of this model and its training data. Because surrogate benchmarks yield a drop-in replacement for raw benchmarks, they enjoy widespread adoption in the HPO community [22, 58, 57, 59–62].

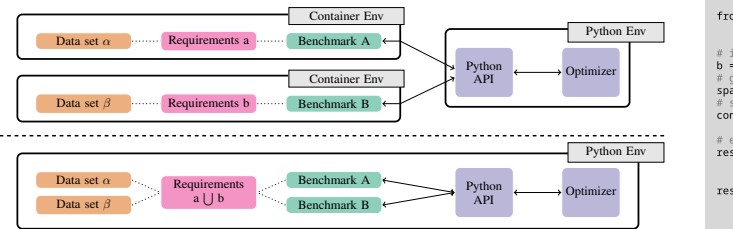

Figure 1: Overview of benchmark environments with (upper) and without (lower) using containers.

```
from hpobench.container.benchmarks.\
    nas.nasbench_101 import NASCifar10ABenchmark

# if necessary downloads container/data
b = NASCifar10ABenchmark(rng=1)
# get hyperparameter space
space = b.get_configuration_space(seed=1)
# sample config at random
config = space.sample_configuration()

# eval at multiple low-fidelities
res = b.objective_function(
            configuration=config,
            fidelity={"budget": 12}, rng=1)
res = b.objective_function(
            configuration=config,
            fidelity={"budget": 108}, rng=1)
```

Figure 2: Code example initializing and evaluating a benchmark.

Furthermore, while *HPOBench* puts a strong focus on multi-fidelity benchmarks, it also facilitates evaluating black-box optimization algorithms. In fact, a multi-fidelity benchmark with $k$ different fidelity levels can be used to define $k$ separate (yet related) benchmarks for black-box optimization. As such, *HPOBench* defines more than $400$ black-box HPO benchmarks.

## 4.3   Reproducibility

One of the challenges that come with many new benchmarks is their one-off development and their lack of maintenance. This means that any new update to the benchmark or its dependencies can easily lead to conflicts and inconsistencies with respect to software dependencies and possibly old published results (see Appendix D.1 for examples). While in practice the very same problem, also known as *dependency hell*, can also occur on the optimizer side, in this paper we focus on the benchmark side.

*HPOBench* circumvents such issues through the containerization of benchmarks using Singularity [63] containers.[3] Each benchmark and its dependencies are packaged as a separate container, which isolates benchmarks from each other and also from the host system. Figure 1 illustrates the advantages that containerization provides, especially when running multiple benchmarks in the same environment. Note that without containers, the environment needs to satisfy the union of all of its benchmarks' requirements (which may actually be mutually exclusive!), while with containers the dependencies for any given benchmark only need to be satisfied once: for the creation of the container. Importantly, the dependencies do not need to be satisfied again for using the benchmarks. Each benchmark is uploaded as a container to a GitLab container registry to provide the history of different versions of the benchmark. Hence, any benchmark created under the *HPOBench* paradigm remains usable without additional bookkeeping or installation overheads for long-term usage. Additionally, no effort is required for maintaining already containerized benchmarks, as long as the API does not change. Although not recommended, each benchmark can also be installed locally along with its specific dependencies without using the containers. We provide a short code sample in Figure 2.

Our notion of *reproducibility* follows the Claerbout/Donoho/Peng convention as summarized by Barba [64]. We preserve benchmarks as containers, so that they can be used without installing all dependencies to obtain the same results. This does not immediately lead to *replicability* on the level of the optimization results. Users need to make sure to for example run a sufficient number of seed replicates to avoid unstable results [65] and to take hardware differences into account when comparing optimizer overhead. Our work differs from other efforts to provide reproducible research. We do not aim to make a single experiment reproducible as *repo2docker* [66] and we also do not aim to package and distribute the whole runtime or workflows as *Jupyter Notebooks* [67] or R's *knittr* [68].

## 4.4   Flexibility

*HPOBench* is a flexible framework that can be used to validate existing HPO research, and develop and improve HPO algorithms, with a focus on multi-fidelity methods. It consists of two sets of benchmarks, which we describe in turn: 22 existing multi-fidelity benchmarks from 7 families that we collected from the multi-fidelity literature (Section 4.4.1); and 88 new benchmarks from 5 families we created to allow a much more flexible use of *HPOBench* (Section 4.4.2).

---

[3]We chose Singularity over the popular Docker (`https://www.docker.com/`) alternative as it (1) does not require super user access and (2) is available on the computer clusters we have access to.

Table 1: Overview of raw (✓), surrogate (✗) and tabular ((✓)) benchmarks. We report the number of benchmarks per family (*#benchs*), the number of continuous (*#cont*), integer (*#int*), categorical (*#cat*), ordinal (*#ord*) hyperparameters and how many are log-scaled. Furthermore, we report the fidelity, the optimization budget and the number of configurations for tabular and surrogate benchmarks.

| Family | #benchs | #cont(log) | #int(log) | #cat | #ord | fidelity | type | opt. budget | #confs | Ref. |
|---|---|---|---|---|---|---|---|---|---|---|
| *Cartpole* | 1 | 4(1) | 3(3) | - | - | repetitions | ✓ | 1d | - | [22] |
| *BNN* | 2 | 3(1) | 2(2) | - | - | samples | ✓ | 1d | - | [22] |
| *Net* | 6 | 5 | 1 | - | - | time | ✗ | 7d | - | [22] |
| *NBHPO* | 4 | - | - | 3 | 6 | epochs | (✓) | $10^7$ sec | 62 208 | [69] |
| *NB101* | 3 | - | - | 26 | - | epochs | (✓) | $10^7$ sec | 423k | [54] |
|  |  | - | - | 14 | - |  |  |  |  |  |
|  |  | 21 | 1 | 5 | - |  |  |  |  |  |
| *NB201* | 3 | - | - | 6 | - | epochs | (✓) | $10^7$ sec | 15 625 | [70] |
| *NB1Shot1* | 3 | - | - | 9 | - | epochs | (✓) | $10^7$ sec | 6 240 | [71] |
|  |  | - | - | 9 | - |  |  |  | 29 160 |  |
|  |  | - | - | 11 | - |  |  |  | 363 648 |  |
| *LogReg* | 20 | 2(2) | - | - | - | iter | ✓, (✓) | 100× | 625 | *new* |
| *SVM* | 20 | 2(2) | - | - | - | data | ✓, (✓) | average | 441 | *new* |
| *RandomForest* | 20 | 1 | 3(2) | - | - | #trees | ✓, (✓) | runtime on | 10k | *new* |
| *XGBoost* | 20 | 3(2) | 1(1) | - | - | #trees | ✓, (✓) | the highest | 10k | *new* |
| *MLP* | 8 | 2(2) | 3(2) | - | - | epochs | ✓, (✓) | fidelity | 30k | *new* |

### 4.4.1 Existing Community Benchmarks

Firstly, to allow comparability with previous experiments, we collected 22 existing multi-fidelity benchmarks from 7 families from the multi-fidelity literature; *HPOBench* preserves these benchmarks by containerizing them and encapsulating them all under a common API (which was not the case before). This not only ensures important previous work to remain accessible, but it also bypasses dependency issues enabling long term usage (see Appendix D.1).

Specifically, these benchmarks comprise raw benchmarks tuning a reinforcement learning agent (PPO on *Cartpole* [22]) and a Bayesian neural network (*BNN* [22]), a random forest-based surrogate benchmark tuning an MLP (*Net* [22]) and four popular NAS benchmark families (*NBHPO* [69], *NB101* [54], *NB201* [70], and *NB1Shot1* [71]). However, these existing community benchmarks also have certain limitations: they are only of limited use for transfer HPO (since there are only between 1 and 6 benchmarks per family), they only offer a single fidelity dimension, and they only evaluate a single metric. We therefore augmented them with 5 new families of benchmarks we describe next.

### 4.4.2 New Benchmarks

To substantially increase the range of possible applications of *HPOBench*, we defined 5 new benchmark families with up to 20 different datasets per family, comprising a total of 88 new multi-fidelity benchmarks. These new benchmarks also provide multiple metrics and multiple fidelity dimensions to go beyond the aforementioned limitations of the community benchmarks.

Our new benchmarks are based on the following popular ML algorithms: **SVM, LogReg, XGBoost, RandomForest**, and **MLP**. All of them evaluate the respective ML algorithm as implemented in scikit-learn [72] and XGBoost [73] on 20 publicly available datasets (8 for the *MLP* due to its high computational cost) from the OpenML AutoML benchmark [74]. We give the OpenML [75] task IDs in Table 7 in Appendix D, which provide fixed train-test splits; for each such task, we used 33% of the training set as the validation split, determined through stratified sampling under a fixed seed. The entire objective function then consists of preprocessing, training the model on the remaining 66% of the fixed OpenML training split, prediction on the fixed validation split, evaluating 4 different metrics (see Appendix D.3), and recording model fit and inference times.[4] The fidelities are algorithm-specific

---

[4]While preparing the CRC we observed that we accidentally trained the models on the OpenML training split. We are currently regenerating the data and will post an updated version of the paper at arXiv:2109.06716.

if possible (number of trees, iterations, epochs) or dataset subsets otherwise (which is used for *SVM*). These benchmarks are available both as raw and tabular versions, have the same API and exist in independent, non-conflicting containers; for the tabular versions, we discretized each hyperparameter (and fidelity) and evaluated 5 different seeds for each configuration of the resulting grid.

Also, four of our new benchmark families (*LogReg*, *RandomForest*, *XGBoost*, *MLP*) allow up to two fidelity dimensions. This enables the development and benchmarking of methods for multi-fidelity optimization with multiple fidelity dimensions, a direction that we deem very promising yet understudied. Similarly, our tabular data collected over multiple datasets (up to 20) allows the effective use of these benchmarks for transfer-HPO, and the recording of multiple evaluation metrics also allows these benchmarks to be used for multi-objective optimization. Moreover, each configuration is recorded on different fidelities with their associated costs, which further lends *HPOBench* great potential in future research in cost-based meta-learning or multi-fidelity multi-objective optimization.

To demonstrate the diversity of our new benchmarks, we show the *empirical cumulative distribution function* (ECDF) for each family in Figure 3. Each line corresponds to one dataset and shows how the objective values are distributed. From the varying amounts of well and badly performing normalized regrets we can conclude that the benchmarks yield different landscapes and thus are diverse in smoothness, resulting in varying algorithm performance. Moreoever, the 5 new spaces vary in their dimensionality (up to 5 for *MLP*), in the hyperparameter data types and their range (see Appendix D).

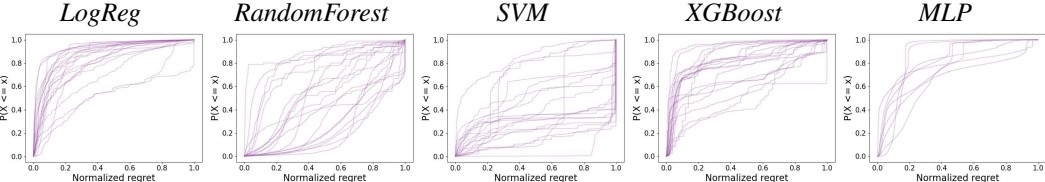

Figure 3: Empirical cumulative distribution. Each plot corresponds to one ML algorithm, and each line within a plot corresponds to one dataset. The lines show the ECDF of the normalized regret of all evaluated configurations of the respective ML algorithm on the respective dataset.

## 5 Experiments

Now, we turn to an exemplary use of our benchmarks in order to demonstrate some features of *HPOBench* and its utility for HPO research. We used our benchmark suite to run a large-scale empirical study comparing 13 optimization methods on our 12 benchmark families (we report detailed results in Appendix H). We first give details on the experimental setup and then study the following two exemplary research questions: **(RQ1)** *Do advanced methods improve over random baselines?* and **(RQ2)** *Do multi-fidelity methods improve over single-fidelity methods?*

### 5.1 Experimental Setup

For each benchmark and optimizer, we conducted 32 repetitions with different seeds to avoid reliance on individual seeds [65]. For our new benchmarks, which have multiple metrics, we minimized $1-$accuracy. For each run, we allowed an optimization budget as described in Table 1 and accumulate time taken by the benchmark (recorded time for tabular benchmarks, predicted time for surrogate benchmarks and wallclock time for raw benchmarks; for our new benchmarks, we used the tabular versions to avoid unnecessary compute costs and $CO_2$ exhaustion) and the optimizer (wallclock time). We kept track of all evaluations and computed trajectories, i.e., the best-seen value at each time step, as follows: If for an evaluation we cannot find another evaluation conducted on the same or a higher fidelity, we treat it as the best-seen value; if it is on the highest fidelity evaluated so far, we treat it as the best seen value if it has a lower loss than the best-seen so far on that fidelity; otherwise, we do not consider this evaluation for the trajectory. This decision reflects the multi-fidelity setting, where a higher budget results in a better estimate of the actual value of interest but can cause jumps in the optimization trajectory, (e.g., when a configuration is the first to be evaluated on a higher budget but is worse than the best configuration on a lower budget). To aggregate and report results, we use either the *final performance* (per benchmark, see Appendix H), *performance-over-time* (per benchmark, see

Appendix H) or *rank-over-time* (across multiple benchmarks). For tabular and surrogate benchmarks we report optimization regret (the difference between the best-found value and the best-known value) and for the other benchmarks, we report the actual optimized objective value.[5]

We give details on the hardware and required compute resources in Appendix E and F and release code for the experiments here: `https://github.com/automl/HPOBenchExperimentUtils`.

## 5.2 Considered Optimizers

We evaluated a wide set of optimizers including baselines for black-box and multi-fidelity optimization. Our selection of optimizers does *not* aim at finding the best optimization algorithm, but to study a broad range of different implementations and tools (for more details see Appendix G). As black-box optimizers, which only access the highest fidelity, we considered random search (*RS*), differential evolution (*DE* [23, 24]) and *BO* with different models: a Gaussian Process model ($BO_{GP}$), a random forest ($BO_{RF}$ [76]), a kernel density estimator (KDE) ($BO_{KDE}$ [22]). Lastly, we also used the winning solution of the NeurIPS BBO challenge (*HEBO* [77]). For multi-fidelity optimization, we used multi-fidelity extensions of some methods mentioned above: Hyperband (*HB* [19]) and its combination with KDE-based *BO* (*BOHB* [22]), with RF-based *BO* (*SMAC-HB* [78]) and with *DE* (*DEHB* [5]). Additionally, we use Dragonfly [79] using a GP with multi-fidelity optimization and combinations of optimization and multi-fidelity algorithms implemented in Optuna [80] (see appendix). [6]

## 5.3 RQ1: Do advanced methods improve over random search?

To demonstrate the validity of our benchmarks, we independently replicate the findings of the 1st NeurIPS Blackbox Optimization challenge [51]: "*decisively showing that BO and similar methods are superior choices over RS and grid search for tuning hyperparameters of ML models*". While this question has already been studied before [14, 81, 36, 15, 33, 82, 83], we will also study it w.r.t. multi-fidelity optimization and using the popular *HB* baseline. We leave out grid search as *RS* has been shown to be superior [81] and as there is no multi-fidelity version of grid search.

We report ranks-over-time in Figure 4, comparing black-box (*DE*, $BO_{GP}$, $BO_{RF}$, *HEBO*, $BO_{KDE}$; 1st column) and multi-fidelity (*BOHB*, *DEHB*, *SMAC-HB*, *DF*; 2nd column) optimizers on *existing community* (top row) and *new* (bottom row) benchmarks. On both benchmark sets most black-box and multi-fidelity optimizers clearly outperform the respective baseline (*RS* (blue) and *HB* (light green)) on average. We also observe that *BO* improves over the evolutionary algorithm *DE* in the beginning, but, except for *HEBO*, looses to it in the very long run on the *existing community* benchmarks [84, 60, 5]. This does not happen on the *new* benchmarks, as their time limits are set more aggressively and the methods developed for this setting (*HEBO* [77], $BO_{GP}$, $BO_{RF}$ and *SMAC-HB* [85]) achieve lower ranks. Considering per-benchmark results (Appendix H), we also observe that methods which appear clearly inferior in the ranking plots perform very well on individual benchmarks (e.g. *DF*[7] on *NB201*). Finally, we find *HEBO* to substantially improve over all other black-box methods.

Besides qualitative measures, we also quantitatively measure whether the advanced methods outperform the respective baselines by counting the number of wins, ties and losses and using the sign test to verify significance [86] on the existing community benchmarks in Table 2 (the new benchmarks yield similar results; see Appendix H). We can observe that four out of five black-box methods are significantly better than *RS*. In the multi-fidelity case, only two out of four methods are significantly better than *HB* and two methods are consistently worse than *HB*. Overall, we conclude that advanced methods consistently outperform random search.

---

[5]Since we study optimizers, we report optimization performance (in the case of ML the validation performance, which is the objective value seen by the optimizer. We note that *HPOBench* in principle allows to compute test performance (the loss computed on a separate test set on the highest fidelity).

[6]We include this framework to show compatibility of *HPOBench* with popular frameworks, but note that it expects to freeze and thaw evaluations. *HPOBench* implements a stateless objective function and, thus, runs that could be thawed and continued instead get accounted the full costs of rerunning them, which slows down optimization. We defer stateful benchmarks to future work.

[7]We would like to note that the bad rank of *DF* for some benchmarks is due to its overhead which prevented it from spending sufficient budget on function evaluations; see Section F for details.

Table 2: P-value of a sign test for the hypothesis that advanced methods outperform the baseline *RS* for black-box optimization and *HB* for multi-fidelity optimization. We underline p-values that are below $\alpha = 0.05$ and boldface p-values that are below $\alpha = 0.05$ after multiple comparison correction (dividing $\alpha$ by the number of comparisons, i.e. 5 and 4; boldface/underlined implies that the advanced method is better than *RS/HB*). We also give the wins/ties/losses of *RS* and *HB* against the challengers.

| | *DE* | $BO_{GP}$ | $BO_{RF}$ | *HEBO* | $BO_{KDE}$ |
|---|---|---|---|---|---|
| p-value against *RS* | **0.00043** | 0.01330 | **0.00001** | **0.00217** | 0.06690 |
| wins/ties/losses against *RS* | 18/2/2 | 15/3/4 | 19/3/0 | 17/2/3 | 13/4/5 |

| | *BOHB* | *DEHB* | *SMAC-HB* | *DF* | |
|---|---|---|---|---|---|
| p-value against *HB* | 0.06690 | **0.00001** | **0.00011** | 0.99783 | |
| wins/ties/losses against *HB* | 13/4/5 | 20/2/0 | 18/3/1 | 5/0/17 | |

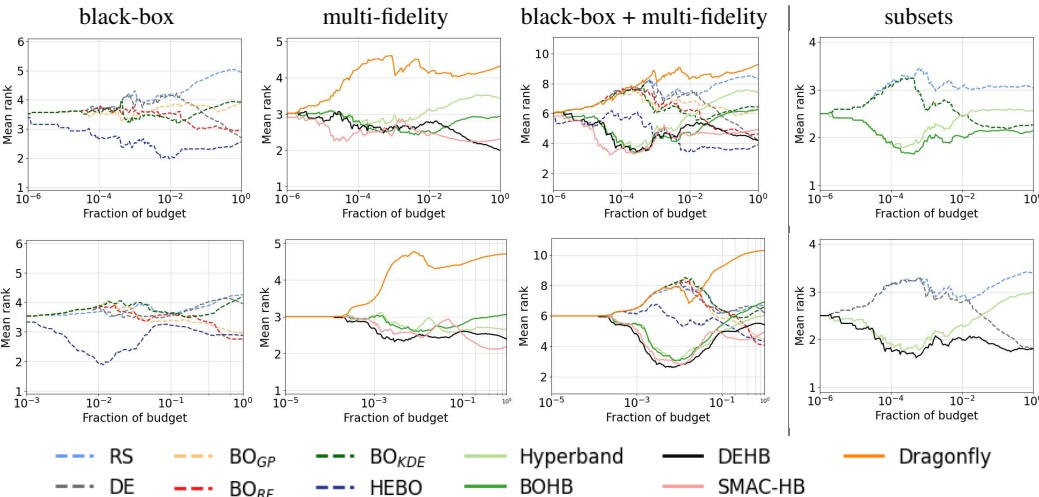

Figure 4: Mean *rank-over-time* across 32 repetitions of different sets of optimizers (lower is better). The left part shows rank across all *existing community* (upper row) and *new* (lower row) benchmarks . The right part reports results on the *existing community* benchmarks only for subsets of optimizers.

## 5.4 RQ2: Do multi-fidelity methods improve over black-box methods?

Next, we study whether multi-fidelity optimization methods are able to consistently improve over black-box optimization methods given a fixed time budget. For this, we look again at ranking-over-time in Figure 4. We first compare black-box methods with their respective multi-fidelity extension, i.e., *DE* vs. *DEHB* and $BO_{KDE}$ vs. *BOHB* in the two plots in the rightmost column. We can see that in the beginning *HB* and the multi-fidelity optimizers perform very similarly and consistently outperform *RS* and the respective black-box version. After a while, the multi-fidelity versions improve over the *HB* baseline, and given enough time, the black-box versions catch up. Second, we compare all optimizers on the *existing community* (3rd column, top) and *new* (3rd column, bottom) benchmarks. Here, we can observe a similar pattern in that *HB* is a very competitive baseline in the beginning but is outperformed first by the advanced multi-fidelity methods and then also by the black-box methods. This is less pronounced on the *new* benchmarks, which we attribute to the tighter time limits.

Similarly to RQ1, we again counted the wins, ties and losses and used the sign test to verify significance [86] on the existing community benchmarks for 100%, 10% and 1% of the total budget in Table 3 (the new benchmarks yield similar results; see Appendix H). We can observe that only *HB* is able to outperform its black-box counterpart for all three budgets we check. For two multi-fidelity methods there is a significant improvement over the black-box methods for 1% of the budget. For the full budget we can no longer state that any of the multi-fidelity methods is statistically better than their counterpart, but judging by the wins and losses the multi-fidelity methods are still competitive.

Overall, multi-fidelity optimizers outperform black-box optimizers for relatively small compute budgets. Given enough budget, black-box optimizers become competitive with their multi-fidelity

versions; in particular, *DE* and $BO_{RF}$ performed very well in the end. However, we need to take into account that for the *existing community* benchmarks the potential catch-up (if at all) only happens after a very substantial amount of (simulated) wallclock time (e.g., 10 Mio. seconds). Hence, multi-fidelity methods are crucial to efficiently tackle real, expensive optimization problems.

Table 3: P-values of a sign test for the hypothesis that multi-fidelity outperform their black-box counterparts. We boldface p-values that are below $\alpha = 0.05$ (implying that multi-fidelity is better).

| Budget | | *HB* vs *RS* | *DEHB* vs *DE* | *BOHB* vs $BO_{KDE}$ | *SMAC-HB* vs $BO_{RF}$ |
|---|---|---|---|---|---|
| 100% | p-values | **0.00074** | 0.73827 | 0.14314 | 0.73827 |
| | w/t/l | 16/5/1 | 6/8/8 | 12/4/6 | 6/8/8 |
| 10% | p-values | **0.00845** | 0.09462 | 0.14314 | 0.50000 |
| | w/t/l | 16/2/4 | 10/9/3 | 12/4/6 | 8/7/7 |
| 1% | p-values | **0.00074** | **0.03918** | 0.06690 | **0.03918** |
| | w/t/l | 17/3/2 | 14/3/5 | 14/2/6 | 13/5/4 |

To conclude, in general when low-fidelities are available and they are representative of the true objective function, multi-fidelity methods are clearly beneficial. In practice, we found that *DEHB* and *SMAC-HB* are reliable multi-fidelity optimizers that work well across the whole collection of benchmarks, while other multi-fidelity optimizers are not able to improve over *HB* consistently. By exploring a very broad range of benchmarks, we also found an existence proof that black-box methods can outperform multi-fidelity methods for very high budgets and that even advanced methods can be outperformed by *RS* in individual benchmarks. We pose it as a challenge to the field to develop methods that do not exhibit poor performance in *any* of the many benchmarks in *HPOBench*.

## 6 Discussion and Future Work

We proposed *HPOBench*, a library for multi-fidelity HPO benchmarks. It serves two purposes: (a) to provide benchmarks with a unified API, and (b) to make them easy to install and use by containerizing them and thus enable rapid prototyping and the development of new multi-fidelity methods that are crucial for ML research and applications. Finally, our library is open-source and we welcome contributions of new benchmarks to keep the library up-to-date and evolve it.

On the technical side, so far, we focused on developing a benchmark library, but we see a large potential in connecting our library with other benchmarking frameworks (e.g. COCO [9] and Bayesmark [8]), optimization frameworks (e.g. Nevergrad [10] and Sherpa [87]) and extending it with further benchmarks [11, 12, 60, 62, 88–90] to increase diversity and to simplify evaluation and comparison of optimizers. For this, it would be interesting to also containerize the optimizers since they can suffer from the same issues as benchmarks. Furthermore, so far, *HPOBench* only contains stateless benchmarks starting a single container. We would like to extend the library to also support optimizers requiring stateful benchmarks (to freeze and thaw evaluations) or running in parallel.

Our set of benchmarks already covers raw, tabular, and surrogate benchmarks, but it would be useful to have all three versions available for all benchmarks, and to automatically generate tabular and surrogate-based benchmarks from raw benchmarks. Also, our new benchmarks can be used to evaluate multi-objective (multiple metrics) and meta-learning (across datasets) methods or even meta-learned multi-fidelity multi-objective methods. We hope for the community to play a large role in defining the protocols for the different special cases; e.g., budgets need to be set differently for black-box multi-objective optimization and single-objective hyperparameter transfer learning. Additionally, it would be interesting to study hierarchical search spaces to cover work in *AutoML*. Furthermore, there is a large potential in automatically creating multi-fidelity benchmarks from any ML algorithm by using data subsets as a low-fidelity.

We also conducted a large-scale study evaluating 13 algorithm implementations to demonstrate compatibility with a wide range of optimization tools, and we thus believe that our library is well suited for future research on multi-fidelity optimization. We showed that advanced HPO methods are preferable over *RS* and *HB* baselines, and that multi-fidelity extensions of popular optimizers improve over their black-box version. Lastly, to reduce computational effort, we would like to study whether we can learn which benchmarks are hard and whether there is a representative subset of them [91].

## Acknowledgments and Disclosure of Funding

We would like to thank Stefan Stäglich and Archit Bansal for code contributions and Stefan Falkner for useful discussions and comments on an early draft of this project. This work has partly been supported by the European Research Council (ERC) under the European Union's Horizon 2020 research and innovation programme under grant no. 716721, and by TAILOR, a project funded by EU Horizon 2020 research and innovation programme under GA No 952215. Robert Bosch GmbH is acknowledged for financial support. The authors also acknowledge support by the state of Baden-Württemberg through bwHPC and the German Research Foundation (DFG) through grant no INST 39/963-1 FUGG.

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
