# OpenReview forum: "HPOBench: A Collection of Reproducible Multi-Fidelity Benchmark Problems for HPO"
_NeurIPS.cc/2021/Track/Datasets_and_Benchmarks/Round2 — NeurIPS 2021 Datasets and Benchmarks Track (Round 2)_

### Official Review · Reviewer_xyi5 · 2021-09-04
**Unclear Contribution**

**Rating:** 7
**Confidence:** 4

**Strengths:**

* The paper organizes many existing benchmarks into a single framework for testing/evaluation.

**Weaknesses:**

* The work does not appear to answer any questions that were previously unknown, or to enable future work to answer those questions.
* The many figures are small, and pixelated, making them hard to read.
* Contributed new benchmarks seem to have no consideration as to why they were selected.

**Additional Feedback:**

Overall I do not see the paper as having a significant contribution. The largest contribution is organizing existing code/benchmarks into a consistent API. I see no discussion of maintenance plan for this framework, and I think it's design of many docker containers and the constantly evolving nature of deep learning frameworks puts it at risk of needing continuous effort to avoid bitrot.

Beyond packaging existing benchmarks, a few basic ML algorithms are included, yet with no more than a paragraph of discussion. Not even an explanation of the kernel being used for the SVM is provided! The results seem to themselves provide no evidence that these new benchmarks provide some information not available from the existing benchmarks, with very similar mean-rank curves in plots. I think a much more thorough and thoughtful discussion of what existing benchmarks do not cover, why that is a problem, and how a expanded set of benchmarks will rectify those issues, would bring this paper to a much greater level of contribution and significance. In it's current form it feels like a bunch of popular ML tools were thrown in "just because" and left at that.


_____

After revision the paper is _much_ improved, uses a proper hypothesis testing approach, and is much easier to read for both new and knowledgable researchers of HPO methods. IMO this work now passes the bar for a NeurIPS paper and am comfortable with acceptance. I do still think the references I provided in initial review would be valuable to fully incorporate, particularly ones like repo2docker that discuss reproducibility with respect to characteristics of this system.

**Clarity:**

The paper is "well written" on a paragraph by paragraph basis, but as a whole is not. I found great difficult understanding what the point/contribution of the paper itself was, and it felt like each paragraph/section had been written independent of any larger theme. I had to get all the way to the first paragraph of the conclusion to feel like I understood what the paper was presenting as it's contribution!

As an example of the writing confusion, I re-read Section 2 several times when first encountered. Each paragraph read as if it was going to explain why one method/type of HPO was better/superior to another, or how some problem was to be fixed, constantly leaving me in expectation of a follow up paragraph that never occurred - instead repeating the process for a new HPO method.

**Correctness:**

The Wilcoxon signed-rank test appears to have been applied incorrectly. It appears the authors in Table 2 are using the test to compare the results of multiple runs as the "rows" in the test, and appear to be performing multiple Wilcoxon tests, one for each row and (winning) column of the table. As both [a,b] note, the proper approach would be that the ranks are obtained per dataset, not per run (one could in this case select the mean score to determine rank) and then apply a non-parameter ANOVA such as the Friedman test to determine significance.

It is also unfortunate that the paper does not take into account it's citation of [13] that the HPO algorithm itself introduces significant noise into the process, as well as the dataset selection. A more robust improvement / tool for benchmarking that accounts for this and looks at the variance of results would have been desirable.

The matter of "reproducibility" in the paper has been a bit too oversimplified. Unfortunately there is inconsistent behaviors on which replication/reproduction means what[k], but many prior works have looked at such "re run the same code" style replication and are not cited/credited [c,d,e,f], and that this is not a panacea and can still lead to replication/reproduction issues is also not discussed/accounted for [g,h].

a. Demšar, J. (2006). Statistical Comparisons of Classifiers over Multiple Data Sets. Journal of Machine Learning Research, 7, 1–30. Retrieved from http://dl.acm.org/citation.cfm?id=1248547.1248548

b. Benavoli, A., Corani, G., & Mangili, F. (2016). Should We Really Use Post-Hoc Tests Based on Mean-Ranks? Journal of Machine Learning Research, 17(5), 1–10. Retrieved from http://jmlr.org/papers/v17/benavoli16a.html

c. Forde, J., Head, T., Holdgraf, C., Panda, Y., Perez, F., Nalvarte, G., … Sundell, E. (2018). Reproducible Research Environments with repo2docker. In Reproducibility in ML Workshop, ICML’18.

d. Kluyver, T., Ragan-Kelley, B., Pérez, F., Granger, B., Bussonnier, M., Frederic, J., … development team, J. (2016). Jupyter Notebooks - a publishing format for reproducible computational workflows. In F. Loizides & B. Scmidt (Eds.), Positioning and Power in Academic Publishing: Players, Agents and Agendas (pp. 87–90). IOS Press. Retrieved from https://eprints.soton.ac.uk/403913/

e. Callahan, B., Proctor, D., Relman, D., Fukuyama, J., & Holmes, S. (2016). REPRODUCIBLE RESEARCH WORKFLOW IN R FOR THE ANALYSIS OF PERSONALIZED HUMAN MICROBIOME DATA. Pacific Symposium on Biocomputing. Pacific Symposium on Biocomputing, 21, 183–194. Retrieved from http://www.ncbi.nlm.nih.gov/pubmed/26776185

f. Claerbout, J. F., & Karrenbach, M. (1992). Electronic documents give reproducible research a new meaning. In SEG Technical Program Expanded Abstracts 1992 (pp. 601–604). Society of Exploration Geophysicists. https://doi.org/10.1190/1.1822162

g. Raff, E. (2019). A Step Toward Quantifying Independently Reproducible Machine Learning Research. In NeurIPS. Retrieved from http://arxiv.org/abs/1909.06674

h. Bouthillier, X., Laurent, C., & Vincent, P. (2019). Unreproducible Research is Reproducible. In K. Chaudhuri & R. Salakhutdinov (Eds.), Proceedings of the 36th International  Conference on Machine Learning (Vol. 97, pp. 725–734). Long Beach, California, USA: PMLR. Retrieved from http://proceedings.mlr.press/v97/bouthillier19a.html

i. Drummond, C. (2009). Replicability is not reproducibility: nor is it good science. In Proceedings of the Evaluation Methods for Machine Learning Workshop at the 26th ICML, Montreal, Canada,2009.

k. Barba, L. A. (2018). Terminologies for Reproducible Research. ArXiv, (January). Retrieved from http://arxiv.org/abs/1802.03311



**Documentation:**

In conjunction with the code the documentation is adequate, though the paper in isolation is not sufficient.

**Ethics:**

I have no ethical concerns.

**Relation To Prior Work:**

See prior Correctness concerns for related work on the methodology side, on the HPO side specifically I see no special issues.

**Summary And Contributions:**

The paper proposes a set of APIs and new benchmarks for Hyper Parameter Optimization (HPO). Running two different classes of HPO algorithms on the dataset they are able to confirm results previously reported in the literature.

---

> ### Author Response · Authors · 2021-09-27
> **Response, Part 1/5**
>
> Dear reviewer,
>
> Thank you very much for your comments. You seem to be an expert on benchmarking ML algorithms and we very much appreciate your feedback, but there appear to be some fundamental misunderstandings of our paper on benchmarking **hyperparameter optimizers**. We will fix all minor issues in an updated version and we now reply to your major points in turn.
>
> Questions regarding Weaknesses
>
> * We respectfully, but strongly, disagree with your statement that “The work does not appear to answer any questions that were previously unknown, or enable future work to answer those questions”. As an example application, our paper answers the research question whether multi-fidelity methods consistently outperform their single-fidelity counterparts, but its much more important contribution is precisely to enable many different lines of future work to answer other questions in HPO. In particular, HPOBench is the first available collection of multi-fidelity HPO problems (and incidentally also the first collection of 100+ blackbox HPO problems, and the first collection of problems that are available both in original form and as tabular versions). Without this collection, papers on multi-fidelity HPO previously had to introduce their own benchmarks (leading to poor comparability across the literature). Furthermore, HPOBench is the first hyperparameter optimization benchmark that is containerized and made available to the community (together with its dependencies) in a way that will still work in a few years from now. Incidentally, HPOBench can also be used for benchmarking transfer learning and multi-objective optimization methods, or any combination of multi-fidelity, multi-objective and transfer learning (all either using the original code or cheap-to-evaluate tabular benchmarks). As such, we strongly believe that it will enable many future works on HPO. We fully expect HPOBench to grow via contributions from the community to answer further research questions that we did not yet have in mind.
> * We apologize for the small figure size; we’ll increase the size in our revision.
> * We give desiderate for the new benchmarks in the fourth and fifth part of our response

---

> > ### Comment · Reviewer_xyi5 · 2021-09-27
> > **Response Response**
> >
> > Thank you for the reply, I'll not cover parts that I feel you fully addressed in rebuttal given its length.
> >
> > >our paper answers the research question whether multi-fidelity methods consistently outperform their single-fidelity counterparts,
> >
> > You have some evidence, but have not performed a Wilcoxon test to show if that result is significant or not and in what ways.
> >
> > >HPOBench is the first available collection of multi-fidelity HPO problems (and incidentally also the first collection of 100+ blackbox HPO problems
> >
> > This is a new claim, I don't think I saw any of that in the paper? I'm now confused about what/where these 100+ problems are.
> >
> > >but to demonstrate that algorithms can be compared to each other based on each benchmark individually.
> >
> > This is not clear in the manuscript, and I'm sure I yet understand the motivation. If they can be compared in individual benchmarks, why do you need many benchmarks? What is the result/value of this approach, and why shouldn't it be that the algorithms are compared across a variety of datasets? The later makes more sense to me, per aforementioned citations.
> >
> > This also does not change that you are doing multiple wilcoxon tests across each row of the table, when it should be a non-parametric ANOVA. As is you are not performing multiple test corrections, and the Table represents hundreds of tests reducing the significance threshold after correction down to ~1e-4! Its also not clear to me why the tests are only done for the old benchmarks and not the new ones.
> >
> > >Would it help if we contrast the methods more against each other, to emphasize their respective pros and cons?
> >
> > I think it would help more if at least major sections had some more text explaining the purpose/how the information will be used. For example, I was not aware that the authors considered the Multi-fidelity vs Black-Box a question being tackled by the paper until your rebuttal. Re-reading I see it is a heading later on, but still I struggle with the fact that the results on old vs new benchmark problems are relatively the same.  Literally stating things like this explicitly & earlier provides valuable context and reduces mental effort to understand the work.
> >
> > >The library will be further maintained by...
> >
> > This should be in a "Datasheets for Datasets" or similar, as the CFP requested.
> >
> > >there have not been any consistently used benchmarks besides artificial test functions such as Branin, Hartman 3 etc. and very simple hyperparameter optimization problems which different teams re-implemented over and over again (2d SVM, 2-layer NN with 4d). Our goal is to provide the community with 1) better benchmarks
> >
> > I think this is the biggest item, which has been elucidated more in rebuttal, but I still feel is lacking. What makes this benchmark better than what existed before, when the results presented seem to show that the new benchmarks result in equivalent conclusions to the prior benchmarks? That we were right, but now more confident, is certainly a valid answer-  but as is I'm not sure what the paper purports to be that answer.
> >
> > Overall I think the author's rebuttal does convince me that there is significant contribution here, but I am hesitant to raise my score primarily due to 1) still existing concerns about hypothesis testing methodology and not using hypothesis testing for RQ1&2. 2) The quality of writing making these details so hard to extract. If the authors can/wish to submit a revision of the manuscript in the window that NeurIPS is allowing I will review and adjust accordingly, though I dislike asking for such a thing due to the stress and time crunch it pushes on people. I understand if the authors do not wish to do so and prefer to argue their points without revision.

---

> > > ### Author Response · Authors · 2021-09-29
> > > **Answer Part 1/2**
> > >
> > > Dear reviewer, thanks a lot for your detailed answer.
> > >
> > > > “You have some evidence, but have not performed a Wilcoxon test to show if that result is significant or not and in what ways.”
> > >
> > > We believe we now fully understand your remarks and give a detailed answer further below.
> > >
> > > > “This is a new claim, I don't think I saw any of that in the paper? I'm now confused about what/where these 100+ problems are.”
> > >
> > > We already highlighted in the abstract that we propose a benchmark collection with more than 100 multi-fidelity problems, which can easily be used as black-box problems when evaluating on the highest fidelity only (which we do in the paper for our black-box optimizers). But we do agree that in our results table these 100+ benchmarks were easy to miss, due to the table format (which had one row for each of the 22 community benchmarks but only one row per new benchmark family; and there are 5 new families with a total of 88 new benchmarks). We made this much more explicit now, thank you for the feedback! We now also explicitly list this as part of our contributions.
> > >
> > > > “This is not clear in the manuscript, and I'm not sure I yet understand the motivation. If they can be compared in individual benchmarks, why do you need many benchmarks? What is the result/value of this approach, and why shouldn't it be that the algorithms are compared across a variety of datasets? The later makes more sense to me, per aforementioned citations.”
> > >
> > > Different benchmarks employ different characteristics, i.e. different search spaces and different optimization landscapes and an optimizer might perform differently well on different types of benchmarks. We collected commonly used benchmarks from the multi-fidelity community to have all under one API. We added the new benchmarks since they allow the community in the future to study one searchspace on many different, but similar tasks (same ML algorithm trained on different datasets). A user can use all or only a subset of these benchmarks, and for our exemplary research questions, we decided to study optimizer performance across all benchmarks (see notes on statistical testing below). We believe that optimization performance aggregated over 100 benchmarks including 12 different search spaces, trained on multiple datasets, offers a good spread of problems such that the results in Section 5 can suggest generally good performing optimizers.
> > >
> > > > “This also does not change that you are doing multiple wilcoxon tests across each row of the table, when it should be a non-parametric ANOVA. As is you are not performing multiple test corrections, and the Table represents hundreds of tests reducing the significance threshold after correction down to ~1e-4! Its also not clear to me why the tests are only done for the old benchmarks and not the new ones.”
> > >
> > > Thank you very much for this comment and further explanation. We applied major changes to the experiment section:
> > >
> > > * We agree that the large results table in its original form did not help answering the exemplary research questions and we thus dropped it (the information is still available in the appendix). We further extended this table by results on the new benchmarks in Table 12-21 in Appendix H.
> > > * We removed the statistical tests from these tables in the appendix based on your preference for not having statistical tests for the individual benchmarks.
> > > * Thank you very much again for your feedback concerning statistical tests for the research questions. Based on this, we now compare RS and HB against other methods (RQ1) and black-box optimizers against their multi-fidelity counterpart (RQ2) across all benchmarks and use statistical tests  following Demsar (2006). For this, we used a sign-test (with multiple testing correction for RQ1 where we compare multiple methods against RS and HB) since we cannot use the Wilcoxon signed-rank test as the different benchmark scores live on different scales.

---

> > > ### Author Response · Authors · 2021-09-29
> > > **Answer Part 2/2**
> > >
> > > > “I think it would help more if at least major sections had some more text explaining the purpose/how the information will be used. For example, I was not aware that the authors considered the Multi-fidelity vs Black-Box a question being tackled by the paper until your rebuttal. Re-reading I see it is a heading later on, but still I struggle with the fact that the results on old vs new benchmark problems are relatively the same. Literally stating things like this explicitly & earlier provides valuable context and reduces mental effort to understand the work.”
> > >
> > > We reworded major parts of the paper to address this comment:
> > >
> > > * We reworded Section 2 to highlight that we do not aim at improving methods, but to give an overview of black-box and multi-fidelity methods which we later use in the study, along with their pros and cons.
> > > * We reworded our contributions at the end of Section 1 explicitly stating all of them including the features of the newly introduced benchmarks
> > > * We reworded large parts of Section 4 (especially Section 4.4) to highlight the necessity of having both, the new and the old benchmarks
> > >
> > > > “This should be in a "Datasheets for Datasets" or similar, as the CFP requested.”
> > >
> > > Our apologies, since the CFP asked for this information only in the case of introducing new datasets. However, we fully agree that a maintenance plan for a benchmark library is necessary and provide this now in the Appendix A.
> > >
> > > > “I think this is the biggest item, which has been elucidated more in rebuttal, but I still feel is lacking. What makes this benchmark better than what existed before, when the results presented seem to show that the new benchmarks result in equivalent conclusions to the prior benchmarks? That we were right, but now more confident, is certainly a valid answer- but as is I'm not sure what the paper purports to be that answer.”
> > >
> > > Thank you for this feedback, we agree that this was not explicit enough in the original submission and reworded Section 4.4 to make this much more explicit. We believe that while the existing benchmarks are great and provide a useful testbed for optimization algorithms, our new benchmarks allow for many more use cases and more thorough experimentation. Just in case this was unclear before, we would also like to note that the community benchmarks were previously not available and their collection and organization under the same interface and containerization is also a contribution of this paper.
> > >
> > > > “Overall I think the author's rebuttal does convince me that there is significant contribution here, but I am hesitant to raise my score primarily due to 1) still existing concerns about hypothesis testing methodology and not using hypothesis testing for RQ1&2. 2) The quality of writing making these details so hard to extract. If the authors can/wish to submit a revision of the manuscript in the window that NeurIPS is allowing I will review and adjust accordingly, though I dislike asking for such a thing due to the stress and time crunch it pushes on people. I understand if the authors do not wish to do so and prefer to argue their points without revision.”
> > >
> > > Thanks for your response and for giving us the opportunity to revise our paper. Regarding 1) Please see our response above about statistical testing methodology and statistical tests for RQ 1&2 (which we now do). Regarding 2) Thank you, we have taken your feedback to heart and carefully revised large parts of the paper to make it much clearer. We hope you agree and are looking forward to your feedback on our latest revision.

---

> > > > ### Comment · Reviewer_xyi5 · 2021-09-29
> > > > **Looking much better & will update after work**
> > > >
> > > > I've taken a quick read over the new Section 4 & 5, it reads _mucher_ better and is far easier to understand then before! I regret I may not have time to fully reply to your rebuttal until quite late this evening due to work, but am impressed with the change & improvement in a short timeframe. I intend to re-read the paper to adjust score properly later this evening, I apologize that it may cut into whenever the authors are(not) awake and rebuttal period ending (not sure if they prevent us from adding comments then).

---

> > > > ### Comment · Reviewer_xyi5 · 2021-09-29
> > > > **Updated Score, Still think related citations should be added if possible.**
> > > >
> > > > The paper is far better in form, methodology, and readability, compared to the first version. I'm comfortable with it being accepted. I do think it would still be improved in the academic sense with some of the citations I provided in initial review, and I would deeply implore the use of vectorized graphics for Figure 4 - its still quite rough to read and would benefit the reader with a computer.
> > > >
> > > > I thank the authors for working with me to figure out what we each needed to know, and hope I did not cause too much stress. I'm impressed at how quickly the writing was improved once we were on the same page.

---

> ### Author Response · Authors · 2021-09-27
> **Response, Part 2/5**
>
> Questions regarding Correctness
>
> “The Wilcoxon signed-rank test appears to have been applied incorrectly. [...]” We think there is a misunderstanding wrt our usage of the Wilcoxon-signed rank test and we hope that we can clarify its use here. [[a](https://www.jmlr.org/papers/volume7/demsar06a/demsar06a.pdf),[b](https://jmlr.org/papers/v17/benavoli16a.html)] aim to compare two or more algorithms across a set of datasets, and [[a](https://www.jmlr.org/papers/volume7/demsar06a/demsar06a.pdf)] suggests to treat datasets (in our case benchmarks) as the entity to compute the differences on. This relates to our work as follows:
> * We expect that there is not one single optimizer that works best on all hyperparameter optimization tasks, but the best-performing optimizer depends on the characteristics of the benchmarks --- a study enabled by HPOBench. Moreover, our goal is not to identify the single best overall hyperparameter optimization algorithm (for that we would need to compare many more implementations), but to demonstrate that algorithms can be compared to each other based on each benchmark individually.
> * Our goal is to quantify the performance of individual algorithms on the individual community benchmarks, taking their variance into account by using repetitions. As we compare the performance of optimization algorithms, we do not know which distribution of outcomes to expect (see 2.1.2 of [Dewancker et al. (2016)](https://proceedings.mlr.press/v64/dewancker_strategy_2016.pdf)). Nonetheless, we can assume the results of seed replicates as defined by [[h](http://proceedings.mlr.press/v97/bouthillier19a.html)] to be commensurable and therefore the Wilcoxon signed-rank test is an appropriate test here.
>
> We are very aware of the tests in [[a](https://www.jmlr.org/papers/volume7/demsar06a/demsar06a.pdf),[b](https://jmlr.org/papers/v17/benavoli16a.html)], and have used the tests proposed by [[a](https://www.jmlr.org/papers/volume7/demsar06a/demsar06a.pdf)] in several publications ourselves in order to answer the question whether one algorithm outperforms others significantly across a set of benchmarks, but in the particular context of HPOBench we believe it is more informative to assess performance differences for each individual benchmark.
>
> “It is also unfortunate that the paper does not take into account it's citation of [[13](https://proceedings.mlsys.org/paper/2021/hash/cfecdb276f634854f3ef915e2e980c31-Abstract.html)] [...]” We think that [[13](https://proceedings.mlsys.org/paper/2021/hash/cfecdb276f634854f3ef915e2e980c31-Abstract.html)] is a great piece of work that demonstrates how one needs to control all sources of variance when conducting machine learning benchmarks. However, we do not compare machine learning algorithms, we compare hyperparameter optimization algorithms. Following the line of arguments from [[13](https://proceedings.mlsys.org/paper/2021/hash/cfecdb276f634854f3ef915e2e980c31-Abstract.html)], our experiments contain two sources of variance: 1) the variance of the HPO algorithm and 2) the variance of the machine learning algorithms to be optimized. To counteract 1), we conduct a total of **32** repetitions. Regarding 2) we pass a different seed to each run at the initialization of the benchmark. Although HPOBench also supports studying noisy benchmarks, we  aim to make them as noise-free as possible (for a given seed) to reduce confounding factors, i.e.  we only measure the variation of the optimizer.
> Different dataset splits would be different variations of the individual benchmarks and would be great to have. However, different dataset splits would be expensive to obtain for tabular benchmarks, as they would increase the cost of obtaining them (NasBench101 for example could only afford 3 seeds for the initialization), and we therefore leave this to future work.
>
> “The matter of "reproducibility" in the paper has been a bit too oversimplified. [...]” Thank you very much for the reference suggestions on reproducibility, we will make sure to cite these and discuss the term “reproducibility” in more detail. Our goal on the lowest level is to preserve benchmarks as containers, so that they can be used without installing all dependencies to obtain the same results. This means we follow the Claerbout/Donoho/Peng convention or “methods reproducibility” following Goodman et al. (following reference [[k](https://arxiv.org/abs/1802.03311)]), which we will clarify in the paper. We agree that a sufficient number of seed replicates need to be done to obtain results that are “inferential reproducible” and we hope that our results are stable as we use 32 replicates to avoid issues as in [[h](http://proceedings.mlr.press/v97/bouthillier19a.html)].

---

> ### Author Response · Authors · 2021-09-27
> **Response Part 3/5**
>
> Questions regarding Clarity
>
> “The paper is "well written" on a paragraph by paragraph basis [...]”
> * We’re sorry to hear that you found our paper to not be coherent beyond the paragraph level. We will improve the flow and especially make sure that our contribution to provide multi-fidelity HPO benchmark problems and a container infrastructure to improve maintainability and usability of the benchmarks becomes clear early on.
> * Regarding the 2nd section, this is a neutral explanation of methods. It’s a bit different from “regular” NeurIPS papers as we do not aim to sell a new method or declare a method the overall best method.  Therefore we do not need to “fix” a previous method. However, we understand your concern. Would it help if we contrast the methods more against each other, to emphasize their respective pros and cons?
>
> Comments in Additional Feedback
>
> “Overall I do not see the paper as having a significant contribution. [...]” We strongly disagree with your statement that the paper does not have a significant contribution. Our motivation for the project stems from our experience developing and benchmarking hyperparameter optimization algorithms over the last 8 years. Throughout the years, there have not been any consistently used benchmarks besides artificial test functions such as Branin, Hartman 3 etc. and very simple hyperparameter optimization problems which different teams re-implemented over and over again (2d SVM, 2-layer NN with 4d). Our goal is to provide the community with 1) better benchmarks and 2) better tools to use benchmarks and make them available to others.
>
> “I see no discussion of maintenance plan for this framework [...]” The library will be further maintained by one of our PhD students. We learned from the painful experience with our first trial on a hyperparameter optimization benchmark in HPOlib [our reference 25, dating back 8 years] and crucially, introduced containers to reduce the required maintenance: with these, we will not have to update benchmarks whenever a dependency of a benchmark is updated. This allows us to focus our maintenance effort on the API and infrastructure itself.
>
> “I think it's design of many docker containers and the constantly evolving nature of deep learning frameworks puts it at risk of needing continuous effort to avoid bitrot.”
> We actually think that the constant nature of evolving DL frameworks is a strong argument **in favor** of our solution, as, in contrast to all existing code-based HPO benchmarks, they are precisely made to avoid the need for constant updating when some dependency changes. E.g., benchmarks such as NasBench101 are already not easily usable with the latest version of Tensorflow, but are still useful benchmarks. We refer to Appendix C.1 for more details on such incompatibilities. Without our tooling it would thus be much harder to use these benchmarks going forward.
> Furthermore, we provide guidance on how to add further benchmarks (see https://github.com/automl/HPOBench/wiki). This will allow efficient contributions by the community. Each new benchmark in a pull request will be properly reviewed by us, ensuring high quality of all benchmarks.

---

> ### Author Response · Authors · 2021-09-27
> **Response Part 4/5**
>
> “Beyond packaging existing benchmarks, a few basic ML algorithms are included, yet with no more than a paragraph of discussion.” We agree that our explanation of the new benchmarks should be extended to explain the reasoning behind them and also present suggested use cases. The set of existing community benchmarks included in this paper are important previous contributions to the literature but are small in size (maximum of 6 datasets) and offer only a single fidelity dimension and only one metric evaluated. One of the major motivations of introducing the new benchmarks was to close the gap that this set of existing community benchmarks leaves:
> * The new set of benchmarks introduce 5 new hyperparameter spaces that are not covered in the curated list of previous benchmarks. Up to 20 standardized datasets were used to collect tabular data on these spaces. Each configuration for each fidelity is evaluated on 5 different seeds for 4 different evaluation metrics. We are not aware of any existing benchmarks for multi-fidelity optimization offering 100+ different benchmarks that are not toy functions, under the same API, designed to run without dependency issues (as everything is packaged in containers and also available as tabular benchmarks). Most importantly, the new benchmarks are expected to make multi-fidelity research easier than it currently is.
> * While none of our curated community benchmarks offer more than one fidelity dimension, four of our new benchmark families (LR, RF, XGB, MLP) allow up to two fidelity dimensions. Thereby, HPOBench can also be used for benchmarking methods for multi-fidelity optimization with multiple fidelity dimensions, a direction that we deem very promising yet understudied -- not least due to the lack of good benchmarks. For the future we aim to provide benchmarks with even more fidelities.
> * Collections of tabular data over multiple datasets open up the new set of benchmarks to be used for transfer-HPO. The recording of multiple metrics also opens these benchmarks for use in multi-objective optimization. Moreover, each configuration is recorded on different fidelities and its associated cost with the possibility of extracting model fit and inference times separately. All of these offer huge potential in future novel research in cost-based meta-learning, multi-fidelity multi-objective optimization, etc. To our knowledge, no other existing benchmarks offer these possibilities within one library. Indeed, we are not aware of **any** other previous benchmark for multi-objective HPO.
> * The new benchmarks included in HPOBench are unique in the aspect that we provide not only the actual benchmark for real evaluations, but also a tabular version over an exhaustively-evaluated grid of hyperparameter values for more efficient experimentation. Both versions share the same API and come with their own containers.

---

> ### Author Response · Authors · 2021-09-27
> **Response Part 5/5**
>
> “Not even an explanation of the kernel being used for the SVM is provided!” We used an RBF kernel and actually did provide this information in Appendix C.2; we will provide the full search space information as well.
>
> “The results seem to themselves provide no evidence that these new benchmarks provide some information not available from the existing benchmarks, with very similar mean-rank curves in plots.” We think this actually speaks in favour of our newly introduced benchmarks, since we can confirm the findings on the existing community benchmarks. The big difference between our new benchmarks and the existing community benchmarks is that they are much more complete than the community benchmarks (the previous community benchmarks are useful for multi-fidelity HPO, but they don’t include multiple metrics, 5 seeds per run, runs across many datasets, etc) and thus allow many new use cases (see the list in Part 4 of our response above.
>
> “I think a much more thorough and thoughtful discussion of what existing benchmarks do not cover, why that is a problem, and how an expanded set of benchmarks will rectify those issues, would bring this paper to a much greater level of contribution and significance.“ Thank you very much for this feedback, we fully agree that the discussion of all our points above and in Part 4 of our response will make the paper much stronger and will include it in the paper.
>
> “In it's current form it feels like a bunch of popular ML tools were thrown in "just because" and left at that.” We used these ML algorithms because they are popular in HPO research and thus relevant for the community ([1](https://jmlr.org/papers/v20/18-444.html),[2](https://dl.acm.org/doi/pdf/10.1145/3449726.3459523),[3](https://2021.ecmlpkdd.org/wp-content/uploads/2021/07/sub_701.pdf),[4](https://arxiv.org/abs/2104.10201),[5](http://www.kdd.org/kdd2018/accepted-papers/view/hyperparameter-importance-across-datasets),[6](https://proceedings.neurips.cc/paper/2018/file/14c879f3f5d8ed93a09f6090d77c2cc3-Paper.pdf),[7](https://proceedings.neurips.cc/paper/2019/file/6ea3f1874b188558fafbab78e8c3a968-Paper.pdf),[8](https://ieeexplore.ieee.org/abstract/document/7344817),[9](http://proceedings.mlr.press/v119/nguyen20d/nguyen20d.pdf),[10](http://proceedings.mlr.press/v80/ru18a/ru18a.pdf),etc). By constructing a benchmark for these ML algorithms we provide a unified implementation to be used in the future. Furthermore, by 1) providing each benchmark on multiple datasets, we hope to bring a more diverse set of HPO benchmarks of popular ML algorithms to be used and 2) by providing them as tabular benchmarks, we, for the first time, also allow very efficient experiments with these.
>
> We thank you for the valuable comments and hope that our clarifications clear up the misunderstandings. We’re standing by to reply to any further concerns you may have.

---

### Official Review · Reviewer_PjUR · 2021-09-18

**Rating:** 6
**Confidence:** 4
**Clarity:** Clear

**Strengths:**

- presents a collection of diverse (multi-fidelity) hyperparameter optimization problems that span traditional HPO problems as well as neural architecture search, in raw/tabular/surrogate form.
- benchmarks are provided as easy-to-use containers
- large-scale evaluation of optimization algorithms

**Weaknesses:**

-  The benchmarks are presented in raw and tabular form, where hyperparameters are discretized on a grid. It would be important to define the range of the grid. An additional column to Table 1 with "# configuration" could be helpful instead of being the appendix.

- It would seem that there is missed opportunity on comparing more advanced black-box methods, e.g. [1],[2], with multi-fidelity methods.

- Another important aspect for HPO benchmarks is transfer learning, which is overlooked in this paper. With increasing research in the community in that direction, the authors can shed some light on the problem and how HPOBench be leveraged towards that end.

References;
[1] Snoek, Jasper, et al. "Scalable bayesian optimization using deep neural networks." International conference on machine learning. PMLR, 2015.
[2] Springenberg, Jost Tobias, et al. "Bayesian optimization with robust Bayesian neural networks." Advances in neural information processing systems 29 (2016): 4134-4142.

**Additional Feedback:**

No

**Correctness:**

There are no details offered about the datasets on the 5 new benchmark families:
- What was the training split used on the data
- What is the reported metric, e.g. cross-validation accuracy, mean squared error , etc. ?
- L227: what exactly is 100x average runtime ?

How many points where used to fit the initial surrogate for single-fidelity BO methods ?

**Documentation:**

Datasets are collected from well-published sources with additional generated dataset. HPOBench is also available as a git repository with easy-to-use containers.

**Relation To Prior Work:**

Authors overlook relevant literature for black-box HPO [3]

[3] Arango, Sebastian Pineda, et al. "HPO-B: A Large-Scale Reproducible Benchmark for Black-Box HPO based on OpenML." arXiv preprint arXiv:2106.06257 (2021).

**Summary And Contributions:**

The paper proposes HPOBench, a collection of 100+ multi-fidelity hyperparameter optimization problems (HPO) from 12 different benchmark families. The authors highlight the importance of low-fidelity approximations compared to the standard black-box optimization problems, with the latter being a specific use-case of the former, and detail some of the existing benchmarks for HPO.  All HPOBench benchmarks are containerized for single task evaluation. In addition to 7 pre-established benchmark families, the authors create 5 new benchmarks and evaluate the performance of 13 optimization methods in a large-scale study, proving once again that advanced methods are better than random search, whereas multi-fidelity algorithms are better than black-box methods with a representative low-fidelity representation of the black-box function.

---

> ### Author Response · Authors · 2021-09-25
> **Response**
>
> Dear reviewer,
> Thank you very much for your feedback and comments. We will answer your questions here now and also update the paper shortly.
>
> Regarding the weaknesses you mentioned:
> * Sure, we will add the number of configurations contained in the tabular benchmarks to Table 1. We will also add more details on the search spaces to the Appendix.
> * We agree that there are many more optimizers which would be interesting to study in the future. In this work we focus on providing benchmarks and describe our library. To demonstrate compatibility with many optimization tools, we evaluate multi-fidelity optimization methods and their black-box versions. We already promised reviewer PETi to look into adding HEBO and think that this is currently the most promising baseline to add, but if time allows we will have a look into adding the reimplementations of [1,2] provided by [RoBO](https://github.com/automl/RoBO/blob/master/robo/fmin/bayesian_optimization.py); if you had any other implementations of those models in mind please let us know.
> * Yes, we agree and also believe that transfer learning is a promising research direction that needs solid benchmarks. With the new benchmarks we aim at closing this gap by providing each benchmark on many different datasets allowing us to study transferring knowledge across datasets. We will add a discussion on this possibility in the paper.
>
> Regarding the questions you raised wrt. Correctness:
> * We used datasets available on OpenML.org which have a fixed training and test split. More concretely, we used OpenML tasks and will provide the task IDs in the appendix. We then used a fixed seed and stratified sampling to split away 33% of the trainingset defined by OpenML as validation data and use the remaining 66% as our actual training data. * Therefore, all train-validation-test splits are fixed for every dataset. We trained the complete pipeline, including preprocessing, on the new training data and computed the objective value to be returned and minimized on the validation set. Thanks for noting that this information is missing, we will add it to the paper.
> * For the new benchmarks, we collect tabular data using 4 different metrics: accuracy, balanced accuracy, f1-score, precision. By default the benchmark API returns the misclassification error as (1 - accuracy), which is what is used for the experiments too. However, the benchmark API offers the option to query other metrics too, enabling future use as multi-objective benchmarks.
> * We consider the average runtime for a specific, tabular benchmark to be the mean runtime in seconds across all configurations collected for that benchmark, evaluated on the highest budget. We then chose the optimization budget for each optimizer at the maximum of either 2 minutes or 100x of the average runtime (to simulate a budget of roughly 100 function evaluations). We will clarify this in line 227.
> * For fitting the initial surrogate for single-fidelity BO tools, we relied on the different tool’s default settings.
>
> Regarding the missing reference: Thanks for pointing this out, we are happy to add this, along with a discussion of how HPObench can be applied to benchmark transfer HPO methods.

---

### Official Review · Reviewer_PETi · 2021-09-20
**Practicable Benchmark for Hyper-parameter Optimisation**

**Rating:** 7
**Confidence:** 4

**Strengths:**

&nbsp;

1. There is a particularly strong effort to ensure both reproducibility and comparability of HPO algorithms in the associated codebase. Particularly, the containerisation paradigm should facilitate HPO algorithm comparison to be performed in the same programming framework.

&nbsp;

2. The codebase for the benchmark is well documented, appears to be actively maintained, and seems to already be garnering interest from the broader HPO community.

&nbsp;

**Weaknesses:**

&nbsp;

## __MAJOR POINTS__

&nbsp;

1. In terms of the codebase documentation, it may be beneficial to include a **representative example of how a user may define a custom algorithm** and benchmark its performance.

2. In the related work section, it may be worth mentioning the Sherpa library [1] as well as the Olympus benchmark for optimisation algorithms [2].

3. It may be worth adding a note to distinguish between multi-fidelity HPO and multi-fidelity BO [3, 4, 5] so that BO researchers don't get confused!

4. It is not abundantly clear to me how the ECDF plots in Figure 3 emphasise the diversity of the new benchmarks. As I understand each line corresponds to a different dataset and hence what the plots show is that a given model's performance varies across datasets. It would be good to state exactly **what kind of diversity is implied** e.g. is the implied conclusion that the black-box functions have differing forms for these tasks?

5. For research question 2 it would be nice if the **nature of the performance improvement could be defined more precisely**. Presumably improvement is meant in terms of wallclock time as opposed to sample efficiency? It would be nice if this was stated explicitly on introducing the question.

6. In answering research question 1 the authors state that, "BO improves over the evolutionary algorithm DE in the beginning but loses to it in the very long run on the existing community benchmarks". In order to support claims such as this, it would be more meaningful to run the BO algorithm that won the NeurIPS 2020 Black-Box optimisation [6](https://github.com/huawei-noah/noah-research/tree/master/BO/HEBO) in place of the standard BO scheme. Otherwise, I would recommend **amending the claim to emphasise that the BO algorithm used is not state-of-the-art and additionally referencing the fact that more performant BO algorithms exist for this purpose [6].**

&nbsp;

## __MINOR POINTS__

&nbsp;

1. It would be nice to have the references in numbered order.

2. Abstract: "To demonstrate the broad compatibility of HPOBench", compatibility with what?

3. The figure axes and font size are quite small rendering them difficult to read. No doubt this can be amended in the camera-ready version for which an extra page is permitted.

4. Reference 51 has no journal. Reference 55, has no website.

5. line 247-249: It may be worth revising this sentence.

6. For Table 2 of the main paper as well as Tables 5 and 6 in the appendix, some row entries only have a single underlined figure. According to the paper the underlined results are not statistically different according to a two-sided Wilcoxon signed-rank test. As such, I would expect either no entries to be underlined or at least two entries per row to be underlined?

7. In Section G of the appendix figures are referenced in non-ascending order.

8. It may be worth supplying the GitHub link in the abstract directly.

9. It is not clear to me what rep. is an abbreviation of in Table 1.

&nbsp;

## __REFERENCES__

&nbsp;

[1] Hertel et al. Sherpa: Robust hyperparameter optimization for machine learning. SoftwareX. 2020.

[2] Hase et al., Olympus: A benchmarking framework for noisy optimization and experiment planning. Machine Learning: Science and Technology. 2021.

[3] Wu et al., Practical multi-fidelity Bayesian optimization for hyperparameter tuning. UAI. 2020.

[4] Song et al., A general framework for multi-fidelity Bayesian optimization with Gaussian processes. AISTATS. 2019.

[5] Kandasamy et al., Multi-fidelity Bayesian optimisation with continuous approximations. ICML. 2017.

[6] Cowen-Rivers et al. An Empirical Study of Assumptions in Bayesian Optimisation, arXiv.  2021.

&nbsp;

**Additional Feedback:**

&nbsp;

All points covered in the main response.

&nbsp;

----------

Update post-rebuttal: The authors have gone to great lengths to address all concerns raised by the reviewers and I have upgraded my score to 7 to reflect the improvements made to the submission!

**Clarity:**

&nbsp;

The paper is relatively easy to follow.

&nbsp;

**Correctness:**

&nbsp;

To the best of my knowledge the claims made by the authors are correct with the exception of a single claim that may warrant amendment as discussed above.

&nbsp;

**Documentation:**

&nbsp;

The codebase is well documented with full details of the collated benchmarks provided in the supplementary material.

&nbsp;

**Ethics:**

&nbsp;

Adequately addressed.

&nbsp;

**Relation To Prior Work:**

&nbsp;

The suggested references to be included are given above.

&nbsp;

**Summary And Contributions:**

&nbsp;

The paper introduces a hyper-parameter optimisation (HPO) benchmark with a particular focus on multi-fidelity HPO. The authors collate 7 existing benchmarks and additionally supplement these with 5 new multi-fidelity benchmarks. Importantly, the benchmarks are containerised and as such, are particularly attractive to practitioners seeking to compare HPO algorithms in the same programming environment.

&nbsp;

I believe the benchmark and associated codebase should serve as a practicable platform to compare HPO algorithms and as such I am leaning towards acceptance. I have a few concerns, which if addressed in the rebuttal, may warrant an increased score.

&nbsp;

---

> ### Author Response · Authors · 2021-09-24
> **Clarification question on references**
>
> Dear reviewer,
> Thank you for your comments and feedback. We are preparing a detailed response, but in the meantime, we would already have one clarification question: In your review you mentioned references [3,4,5] for distinguishing between multi-fidelity HPO and multi-fidelity BO. However, it seems like the references for these are missing. Could you please give more details, so we can consider them for our response?

---

> > ### Comment · Reviewer_PETi · 2021-09-24
> > **References Added!**
> >
> > &nbsp;
> >
> > Many thanks for pointing this out! References now added in the main review!
> >
> > &nbsp;

---

> ### Author Response · Authors · 2021-09-25
> **Response**
>
> Dear reviewer,
>
> Thank you very much for your comments and mentioning that you may further increase your score. We will fix all minor issues in an updated version shortly and reply to your major points now:
>
> 1. Custom Algorithms: The goal of our library is to first define benchmarks and allow users great flexibility in how exactly to use it. For example, we support blackbox optimization, multi-fidelity optimization, hyperparameter transfer learning (across the 20 datasets for our new benchmarks), optimization of multiple metrics, and arbitrary combinations of these. Users can choose the most appropriate subset of benchmarks and budgets, and also whether to use original benchmarks or tabular surrogates. We already have [a few examples](https://github.com/automl/HPOBench/tree/master/examples/w_optimizer), which we will update and extend shortly with an example for a blackbox optimizer, but the sky is the limit. We agree that also having a structured framework available to set up comparisons for various special cases would be beneficial; we will extend the discussion on this in the future work section.
> 2. Thanks a lot, we will add these shortly.
> 3. We distinguish multi-fidelity HPO and multi-fidelity BO exactly like we distinguish HPO and BO. HPO is a problem and BO is a method for solving it. Multi-fidelity HPO methods include multi-fidelity BO methods but also, e.g., Hyperband and multi-fidelity evolutionary methods, such as DEHB. We will clarify this in the text.
> 4. Yes, exactly, each line corresponds to one dataset and shows how the objective values are distributed. On the one hand this allows us to conclude the performance of random search, which varies across datasets. On the other hand this also means that the black-box functions have different error landscapes because there are different amounts of well and badly performing configurations. With such different characteristics, it is likely that the performance of other algorithms also varies across datasets. We will add a further explanation to the paper.
> 5. Thanks a lot for this comment. Indeed, we meant the final performance and performance over time wrt wall-clock time and will add this to the text.
> 6. Thanks, we agree that we should clarify that there are more advanced and better BO methods available. We will add more details to our paper and especially emphasize that there are better and more state-of-the-art algorithms. Furthermore, we aim to include HEBO into our comparison and will keep you posted wrt to progress in this direction.
>
> Regarding your minor points: We will add and clarify all points you mentioned, i.e. changing the order of references, add more details to the abstract, increase figure size in the main paper, fix references and add more details to Table 1.
>
> Regarding the question on “broad compatibility”: We meant compatibility with optimization tools and will clarify this.
>
> Regarding the Wilcoxon signed rank test: We underline results if they are not statistically different from the **best** result (and always underline the best result, too) and we will clarify this in the table caption.

---

> > ### Comment · Reviewer_PETi · 2021-09-25
> > **Thanks for the Clarifications!**
> >
> > &nbsp;
> >
> > 1. Many thanks for pointing towards the cartpole examples! It does indeed make a lot of sense that the implementation details of a new HPO algorithm will depend on how the user has defined it and so the template examples using Hyperband and BOHB are quite possibly the most appropriate guides! I consider this point resolved!
> >
> > 2. Excellent!
> >
> > 3. This is an interesting point. As I understand, the multi-fidelity HPO case concerns situations in which approximations are made to the underlying problem to enable faster training of HPO algorithms. The authors are then stating that this problem definition maps onto multi-fidelity BO in which the black-box can be queried at multiple fidelities? That makes a lot of sense!
> >
> > 4. Thanks for the clarification on this point! The notion that the ECDF plots support the fact that the HPO problems have optimisation landscapes of varying smoothness makes sense.
> >
> > 5. Excellent!
> >
> > 6. I look forward to seeing the results! I understand that a week is not a lot of time to implement another baseline and so I will naturally bear this in mind!
> >
> > &nbsp;
> >
> > Many thanks for the clarifications on the table results. The underlining scheme now makes sense! I look forward to seeing the revised manuscript and will increase my score with the suggested amendments incorporated!
> >
> > &nbsp;

---

### Official Review · Reviewer_95z4 · 2021-09-20
**HPOBench: A Collection of Reproducible Multi-Fidelity Benchmark Problems for HPO**

**Rating:** 7
**Confidence:** 3
**Correctness:** Yes
**Clarity:** The paper is mostly clear. Some point…

**Strengths:**

•	Unified API for different tasks
•	Containerization for reproducibility
•	Collection of diverse tasks
•	Detailed experiments of different methods


**Weaknesses:**

•	The definition 1 of HPO benchmark should be more clarified.
•	Table 1 is a bit hard to follow. Kindly consider a better format
•	As by empirical experiments, it seems current algorithms from the community are good enough on this benchmark and there is limited space to improve, thus the value of the benchmark is doubted. Kindly consider the value of the benchmark itself.


**Additional Feedback:**

No additional feedback

**Documentation:**

The documentation on Github is clear.

**Relation To Prior Work:**

The authors discuss previous benchmark attempts for HPO.

**Summary And Contributions:**

This paper proposes a benchmark for multi-fidelity HPO including a diverse set of tasks (different search space, surrogate, previous work, and new tasks). This benchmark alleviates the issue of reproducibility, open-source, consistency as in previous efforts. An empirical comparison of different HPO algorithms is conducted and authors ensure that advanced methods outperform random search and that multi-fidelity algorithms are beneficial.

---

> ### Author Response · Authors · 2021-09-25
> **Response**
>
> Dear reviewer, thank you very much for your comments and your positive feedback.
>
> We will add more details to Definition 1 and reformat Table 1 shortly, thanks for pointing these out.
>
> Regarding value of the benchmarks: While many optimizers eventually obtain good performance, the overall budget required for this is still large for many optimizers and benchmarks. We believe there is a lot of space left for improvement, e.g. there could be methods that leverage transfer learning (which our new benchmarks support, having been built on 20 datasets), make better use of lower fidelities, make sure that multi-fidelity optimizers perform en-par or better than the respective black-box version in the long run, reduce optimizer overhead or work better in some other form (e.g. using other optimization techniques such as local search or learning the optimizer). With HPOBench we provide a basis for developing and comparing such methods.

---

### Author Response · Authors · 2021-09-29
**Revised PDF and Appendix available**

Dear reviewers,

Thank you again very much for your time and feedback. We have uploaded a revision of our paper, taking your feedback into account. We addressed all comments and applied major changes to Section 1, 2 and 4. We are still working on integrating HEBO (PETi) and a paragraph on the terminology of reproducibility (xyi5), but in the meantime we would appreciate feedback on whether our revision addresses the issues raised in the reviews.

Yours sincerely, the authors

---

### Author Response · Authors · 2021-09-30
**Further updates**

Dear reviewers,

@Reviewer #4: Thank you very much. We appreciate that you updated your score. Yes, we will definitely add a paragraph on reproducibility based on the references you provided for our next revision.

@Reviewer #2: We have good news regarding HEBO. We have just finished integrating it in our [experiment code](https://github.com/automl/HPOBenchExperimentUtils/pull/54/files) and it ran fine on a few initial tests. We are currently running larger experiments and if these results look fine as well, we are optimistic that we can add them to the paper.

Yours sincerely, the authors

---

> ### Comment · Reviewer_PETi · 2021-09-30
> **Many Thanks for the Additional Implementation; Upgraded Score**
>
> &nbsp;
>
> Many thanks for including the HEBO implementation! Following the discussions with reviewer 4 I am greatly impressed by the commitment from both parties to engage in a constructive dialogue. The authors have gone to great lengths to address all reviewer concerns including statistical hypothesis tests, additional implementations and rewriting of the manuscript. I commend the authors for their efforts and raise my score accordingly!
>
> &nbsp;

---

### Decision · Program_Chairs · 2021-10-09

**Decision:**

Accept

**Comment:**

All reviewers agree on acceptance.